# Evaluation of the Emulsification Properties of Marine-Derived Rhamnolipids for Encapsulation: A Comparison with Commercial Surfactants

**DOI:** 10.3390/biom15101451

**Published:** 2025-10-14

**Authors:** Sara Gorrieri, Carmine Buonocore, Giulia Donà, Chiara Pezzoli, Martina Vakarelova, Daniela Coppola, Fortunato Palma Esposito, Donatella de Pascale, Gerardo Della Sala, Francesca Zanoni, Pietro Tedesco

**Affiliations:** 1Department of Biotechnology, University of Verona, Strada le Grazie 15, 37134 Verona, Italy; sara.gorrieri@univr.it; 2Department of Ecosustainable Marine Biotechnology, Stazione Zoologica Anton Dohrn, Via Ammiraglio Acton, 55, 80133 Naples, Italy; carmine.buonocore@szn.it (C.B.); daniela.coppola@szn.it (D.C.); fortunato.palmaesposito@szn.it (F.P.E.); donatella.depascale@szn.it (D.d.P.); gerardo.dellasala@szn.it (G.D.S.); 3Sphera Encapsulation S.r.l, Via Alessandro Volta 15A, 37062 Villafranca di Verona, Italy; dona@spheraencapsulation.com (G.D.); vakarelova@spheraencapsulation.com (M.V.); 4Department of Civil Chemical Environmental and Materials Engineering, University of Genoa, Via Montallegro 1, 16145 Genova, Italy; chiara.pezzoli@edu.unige.it

**Keywords:** rhamnolipids, glycolipids, biosurfactants, encapsulation, micelles, coenzyme Q10

## Abstract

Rhamnolipids are a class of glycolipids known for their surface and emulsifying activity. These molecules, produced by a few Gram-negative genera, mostly Pseudomonas, offer natural alternatives to synthetic surfactants in different industrial fields. This study examines the emulsifying and encapsulation performance of Rhamnolipids derived from the marine Antarctic bacterium *Pseudomonas gessardii* M15, comparing its emulsification ability and stability with those of commercial surfactants, Sodium dodecyl sulfate (SDS) and sucrose esters (SE), under extreme conditions of temperature and pH. The Rhamolipids were used to encapsulate Coenzyme Q10 with Arabic gum as the carrier matrix. Rhamnolipids exhibited surface and emulsifying activity comparable to that of SDS and superior to SE at neutral and basic pH levels. Their performance declined under acidic conditions, whereas exposure to 90 °C had no significant effects. The encapsulation efficiency of Coenzyme Q10 was significantly higher in the case of Rhamnolipids, with a percentage of encapsulated compound of 99.6 ± 0.2%, compared to the 38.2 ± 7.1% found when SDS was used. Rhamnolipids extracted from *Pseudomonas gessardii* M15 exhibit strong potential as a natural surfactant, particularly in formulations that require thermal stability and effective encapsulation. These findings support its use as a sustainable alternative to synthetic agents in diverse industrial settings.

## 1. Introduction

Surfactants are a class of molecules used in various sectors, including the cleaning, food, agrochemical, and pharmaceutical industries, with a growing demand. These are a broad group of molecules characterized by an amphiphilic nature, which gives them the capacity to position at interfaces (water/oil and water/air), stabilizing them [1,2]. Therefore, these molecules are employed to create stable emulsions, which form the basis for most encapsulation technologies (e.g., spray drying, ionic gelation, and coacervation) that aim to enclose a molecule or a group of molecules of interest within a protective membrane [3,4]. Surfactants can be divided into three groups: synthetic surfactants, bio-based surfactants, and biosurfactants (BSs). The first ones are obtained from petrochemical sources and represent the majority of currently used surfactants, reportedly having the highest environmental impact. Bio-based surfactants are produced using more sustainable sources, such as vegetable oil (e.g., palm and corn oil). However, they are often reported to be allergenic, have low biodegradability, and, nonetheless, compete with humans for food resources [2,5]. BSs are produced from microbial strains as secondary metabolites and are recognized as more biodegradable and less toxic than their synthetic counterparts. They also exhibit interesting bioactivities, such as antioxidants, antiviral, and antibacterial properties [6]. Thanks to these properties, there is great interest in the broader utilization of BSs in many applications as substitutes for synthetic ones. Rhamnolipids (RLs) are among the best-studied and industrially relevant BSs, showing excellent surfactant/emulsifier properties comparable to their artificial counterparts [1]. RLs are glycolipids produced by Gram-negative strains. RLs consist of one or two molecules of L-rhamnose linked to one or two β-hydroxy fatty acids [7]. The fatty acid chains can be saturated or unsaturated, with lengths ranging from 8 to 24 carbon atoms; however, fatty acids with 8–14 carbon atoms are the most common. The properties of RLs primarily arise from the type of microbial strain, fermentation conditions, and the substrates used in their production. To date, more than 60 different RL congeners have been reported, differing in the length of the fatty acid chain and in the degree of unsaturation [8]. *Pseudomonas aeruginosa* is the primary producer of RL, although other bacterial genera, such as *Nocardiopsis, Acinetobacter, Enterobacter,* and *Burkholderia*, have also been described as RL producers. [9]. Thanks to their amphiphilic structure, RLs exhibit remarkable surface-active properties, reducing surface tension to 30–32 mN/m, and are therefore considered promising candidates to replace or complement synthetic surfactants in several industrial sectors. Beyond their surface activity, RLs possess diverse bioactivities, including anticancer, antifungal, antibacterial, antioxidant, antibiofilm, and antiviral effects, that make them promising candidates for applications in biomedicine, agriculture, environmental remediation, and food processing. They also enhance product safety by reducing microbial contamination and oxidative degradation, contributing to extended shelf life. The market value of biosurfactant is expected to reach about $5.99 billion USD by 2029 [10], and there are already examples of RLs used as additives in commercial cosmeceutical formulations produced by Evonik and TeeGene. This highlights their potential for further industrial development and innovation motivating ongoing studies in fermentation and downstream process optimization [9]. The most studied RLs are obtained from *Pseudomonas aeruginosa* strains, a human opportunistic pathogen [8]. Previous studies have characterized RLs from *P. aeruginosa* for applications such as oil recovery, where their emulsifying capacity and interfacial activity played a crucial role [11]. In a comprehensive study, rhamnolipids were characterized with respect to their structural variations for emulsifying activity and particle formation. Interestingly, the number of rhamnose residues had little effect on the average particle size; however, nanoparticles containing disaccharides exhibited lower polydispersity index (PDI) values compared to monosaccharide-containing RLs, suggesting that a larger hydrophilic head group favored a narrower size distribution. Shorter-chain rhamnolipids produced the smallest particles with very low PDI values, making them suitable for drug-loading applications [12]. RLs have also been explored for encapsulation in both nutraceutical and pharmaceutical contexts. For instance, curcumin, essential oil and doxorubicin have been successfully encapsulated using RLs in combination with other polymers [13,14,15] highlighting their potential as versatile carriers.

Despite some *P. aeruginosa* RLs mixture being already well characterized and commercially available, the Antarctic marine bacterium *Pseudomonas gessardii* has received little attention regarding its potential applications. Preliminary studies have shown that *P. gessardii* can produce an RLs mixture with compositions and biocidal activities distinct from those of *P. aeruginosa* [9]. However, few studies have investigated the use of *Pseudomonas gessardii* RLs in applied contexts, particularly in encapsulation for biotechnological or industrial purposes. Encapsulation is a technique that enables the incorporation of active compounds within protective carriers, allowing their controlled release under specific physicochemical and biological conditions. Biopolymers are widely used as coating materials due to their favorable properties, including biodegradability, stability, and nontoxicity. Combining biosurfactants with biopolymers in encapsulation systems could enhance the biological activity, physicochemical stability, and shelf life of products, making this approach highly promising for food, biomedical, and industrial applications. Therefore, further studies are necessary to fully characterize *P. gessardii* and assess its suitability for RL-based encapsulation strategies. Additionally, only a limited number of studies have demonstrated the efficacy of biosurfactants in systems similar to those used in industry. In this study, we focused on evaluating the emulsifying capability of RLs from *P. gessardii* M15 in the form of a crude extract (M15CE) and a purified mixture (M15RL) under different pH and temperature conditions to assess the potential use of M15RL as an encapsulation agent. Specifically, we compared the surface tension, critical micelle concentration (CMC), and emulsion index of M15RL with those of selected commercial surfactants to evaluate their potential for oil encapsulation. Sodium dodecyl sulfate (SDS) and sucrose esters were included as representative commercial surfactants. SDS is an anionic surfactant composed of a 12-carbon alkyl chain linked to a sulfate group, widely used in industrial and laboratory applications, as a detergent in personal hygiene products such as toothpaste or shampoo, or as an extraction tool for protein and DNA [16]. Sucrose esters are plant-derived nonionic surfactants in which one or more fatty acids are esterified to a sucrose molecule; they are biocompatible and widely used in food and cosmetic formulations due to their mildness, safety, and emulsifying capacity [17]. The choice of these two surfactants was not intended for a structural comparison, but rather for a functional one. Specifically, SDS represents a widely used synthetic surfactant, while sucrose esters are a well-established example of natural surfactants. Our aim was therefore to compare their performance in emulsification and encapsulation with that of the newly investigated microbial biosurfactant, which remains relatively unexplored. Furthermore, a proof of concept (PoC) encapsulation study was conducted using the lipophilic active ingredient Coenzyme Q10 (Q10), demonstrating the high value of RLs in such context.

## 2. Results

### 2.1. Chemical Characterization of Rhamnolipids Mixtures

The M15CE and M15RL, obtained after solid-phase-extraction (SPE) fractionation of the crude extract, were analyzed through Liquid Chromatography—High-Resolution Tandem Mass Spectrometry (LC-HRMS^2^). The mass spectrometry analysis allowed the identification of RLs congeners present in the two mixtures, as reported in Table 1.

The data showed that the two RL mixtures are quite similar, except for the absence of mono-rhamno-mono-lipids, which eluted from minutes 16 to 24. In addition, SPE purification increased the relative abundances of RLs congeners eluted in the middle part of the chromatogram, from minute 27 to 35, while reducing the concentration of the others. However, the most abundant RLs in M15CE, Rha-C12:1-C10, also represented the most abundant in the purified mixture M15RL, accounting for 29.2% and 27.4% of the two mixtures, respectively. The only exception was represented by Rha-C8-C10, which decreased from 10% in M15CE to 2% in M15RL. The most abundant RLs in both mixtures were Rha-C10-C10, Rha-C10-C12:1, and Rha-C12-C10, which combined accounted for 70% of the total rhamnolipids, in accordance with our previous results [9]. *P. gessardii* RLs mixtures are entirely made of Mono-RLs, unlike other Pseudomonas species (e.g., *P. aeruginosa*), due to the lack of the *rhlC* gene responsible for adding the second rhamnose unit. This represents a great advantage as mono-RL mixtures present antimicrobial activities and enhanced emulsification performance [18,19]. Moreover, the higher proportion of Mono-RLs correlates with lower critical micelle concentration (CMC) values of RL mixtures [20]. M15CE and M15RL showed solubility in demineralized water without aggregate formation. Water solubility reflects the HLB of M15RL, which was assessed to a value close to 13, thus indicating the ability of the mixture to stabilize *o*/*w* emulsion [21].

### 2.2. Evaluation of Surface Tension and Critical Micelle Concentration

Surface tension vs. concentration plots (adsorption isotherms) are reported in the Appendix A (Appendix A). These data were used to estimate the CMC values and to guide formulation selection. Both M15CE and M15RL showed improved surfactant activities compared to commercial surfactants. Indeed, their CMC values were 0.024 ± 0.006 mg/mL and 0.015 ± 0.001 mg/mL, respectively, approximately 7 times lower than those of SDS (0.142 ± 0.020 mg/mL) and SE (0.139 ± 0.005 mg/mL). The absorption isotherms show that, in the case of M15CE and M15RL, ST clearly decreases with increasing surfactant concentration, indicating effective adsorption at the air–water interface (Appendix A). A similar trend was observed for SDS (Appendix A), although the curve exhibited more fluctuations, likely due to experimental sensitivity or the ionic nature of the surfactant. In contrast, SE showed a minor decrease in surface tension which may indicate a lower interfacial activity (Appendix A). In the case of SDS, the measured CMC differs from values commonly reported in the literature (≈8.2 mM, ≈2 mg/mL) [22]. However, it is well established that CMC determinations are highly sensitive to experimental conditions, including the measurement technique, water purity, and temperature. In our experiments, surface tension decreased from 60.79 ± 0.64 mN/m at 0.001 mg/mL to a plateau of ~30 mN/m at 0.15 mg/mL, as shown in the isotherms reported in Appendix A. Therefore, the CMC value obtained in our experiments reflects the specific experimental conditions and remains consistent with the behavior reported for SDS. On the other hand, the CMC value of M15RL was comparable to that reported in the literature (0.013 mg/mL) [23]. Meanwhile, the slightly higher CMC value of M15CE was hypothesized to be related, both to the reduced concentration of RLs in the crude extract and to the differences in the RLs composition, such as the presence of unsaturated bonds and the length of the aliphatic chains [20,24].

### 2.3. Surface Activity at Different pH

To evaluate the performance of RLs as a possible encapsulation aid, it is essential to understand their performance under various conditions expected in the food and cosmetic fields, such as high temperatures and basic and acidic pH levels [19]. To achieve the CMC of all the tested samples and mimic the concentration used in the industry, 0.2 mg/mL was selected. The samples were subjected to different temperature and pH conditions, with surface tension (ST), particle size, and zeta potential chosen as indicators of surface activity. Figure 1 shows the results for micelle size, charge, and surface tension as a function of the pH.

At neutral pH, the rearrangement of SE molecules led to the formation of the smallest aggregates (120.20 ± 0.70 nm), followed by M15CE (155.10 ± 3.80 nm), M15RL (166.10 ± 8.20 nm), and SDS (220.90 ± 79.90 nm), respectively. All the samples exhibited a negative surface charge, ranging from −23.40 to −49.70 mV, with both M15RL and M15CE having the most negative values (−49.70 ± 3.70 and −46.10 ± 4.10 mV, respectively). The high charge of RLs is due to the carboxylic acid groups, which are negatively charged at neutral pH [3]. At an acidic pH, the micelles produced by M15CE showed a non-significant increase (160.90 ± 3.40 nm) and a less negative surface charge value (−29 ± 1.70 mV). In contrast, a significant rise in micelle diameter was observed in M15RL (272.30 ± 13.60 nm), accompanied by a slight decrease in the absolute value of surface charge to –27.70 ± 0.20 mV. The lowering effect on the absolute value of the surface charge of RLs at acidic pH has already been reported due to the proximity to their pKa, where the non-ionized form of RLs tends to precipitate, losing their activity. The repulsive force observed in the case of micelles formed in solution was still considered sufficient to resist aggregation forces [25]. On the other hand, the acidic pH had a significant impact on the surface activity of SDS and SE. Indeed, they formed bigger aggregates, e.g., 421.60 ± 58.50 nm in the case of SDS and 5371 ± 1212 nm for SE. The observed pH-dependent behavior of SE aggregates is consistent with previous findings reported in the literature, which also demonstrated structural changes in sucrose ester systems upon variation in pH [26]. The acidic pH also induced a significant decrease in the absolute value of the surface charge, with −12.70 ± 0.10 mV in the case of SDS and −2.20 ± 0.70 mV in the case of SE. Although sucrose esters are nonionic surfactants and do not inherently carry a charge, the measured Z-potential values can be explained by secondary effects such as preferential adsorption of hydroxyl ions at the aggregate surface, partial hydrolysis of ester groups, or the presence of trace ionic impurities in the aqueous medium. On the other hand, a minor variation was observed in the ST of both SDS and SE, at 49.30 ± 0.40 mN/m and 35.80 ± 0.80 nm, respectively. The slightly negative value of the surface charge of SE underlined that the conditions were close to its pKa, with a value of a zero net charge. At pH 12, the surface activity was restored to values close to those registered at neutral pH with −53.40 ± 1.40 mV for M15CE and −49.70 ± 3.59 mV for M15RL. The same trend was also observed in the case of SDS (−31.70 ± 3.21 mV) and SE (−37.80 ± 5.90 mV). Both surfactants also registered a significant increase in the size of the micelle, 352 ± 34.10 nm (*p*-value = 0.007) and 512.90 ± 62.90 nm (*p*-value = 0.012), respectively. The increase in particle size from pH 7 to pH 12 was not significant for both M15CE and M15RL. Overall, it was observed that M15CE and M15RL have higher stability than the other tested surfactants at different pH levels.

LC-MSMS analysis of the treated mixtures revealed no significant differences compared to the control samples. The total amount and composition of RLs remained stable with few exceptions. A trend was observed in the decrease in the relative abundances of Rha-C8-C10 and Rha-C10-C10, which paralleled the increase in the amounts of Rha-C12:1-C10 and Rha-C12-C10 in both M15CE and M15RL at pH 12. However, these variations in the relative amounts of RL congeners were in the 2–5% range and did not significantly affect the total composition of both mixtures (Appendix A). Fatty acid rearrangements in the RL structures have already been reported to occur at alkaline pH values [27]. These analyses strengthen the hypothesis that the differences in the Z-potential were primarily due to the net charge of the RLs in alkaline conditions, rather than degradation reactions of the RLs. Overall, both crude and purified RLs showed remarkable stability against pH variations, as indicated by micelle sizes that remained stable across the tested range and by high surface charge values, supporting their resistance to aggregation. Conversely, SDS, being an anionic surfactant, exhibited pH-dependent fluctuations in micelle size, likely related to changes in electrostatic repulsion and ionic strength effects. At acidic conditions, partial neutralization of charges may have reduced electrostatic stabilization, favoring the formation of larger clusters or aggregates. The most evident destabilization was observed in the case of SE at pH 3, where the solution turned from clear to opaque, with visible aggregates corresponding to the large micelle sizes and a zeta potential approaching zero. Although SE is considered a nonionic surfactant, this unexpected behavior may derive from partial protonation of residual fatty acid moieties or altered hydrogen bonding at low pH, which can reduce colloidal stability. Collectively, the absence of significant variations in micelle size across the tested pH values, together with mass spectrometry analysis showing no structural changes in RLs, highlights their stability under pH conditions often encountered in industrial applications. This property enables RLs to outperform SE in terms of stability, despite their micelles being larger in size.

### 2.4. Thermal Treatment

The influence of thermal treatment (TT) on the surface activity was tested at 90 °C (Figure 2).

The data showed that the average size of micelles was poorly affected in both cases of M15CE and M15RL. In contrast, the surface charge increased to −54.60 ± 0.30 mV for M15CE and to −62.20 ± 1.20 mV for M15RL. A destabilization phenomenon was observed in the SDS sample after treatment with TT. This conclusion was suggested by the increase in the average diameter of its micelles (516.40 ± 262.50 nm) and the poor quality of the data obtained from the DLS (PDI value > 0.5). It was hypothesized that SDS may exhibit temperature-dependent phase behavior, as observed in concentrated solutions (e.g., 10% *w*/*w*), which can solidify below ~20 °C, indicating that micelle aggregation is highly sensitive to temperature. In our experiments, heating likely induced transient structural rearrangements, promoting the association of micelles into larger clusters, as detected by DLS.

In the case of SE, a non-significant decrease in micelle size and surface charge (−29.50 ± 1.67 mV) was observed. It can be argued that the better performance of SE is due to its enhanced solubility after TT, since a water temperature of 80 °C is required for its solubilization. Interestingly, Pagureva et al. also reported a temperature-dependent change in the appearance of sucrose ester solutions, with a transition from an opaque dispersion at 25 °C to a transparent solution at 50 °C, accompanied by the formation of spherical droplets observable under the microscope [17]. This finding is consistent with our results, which indicate that temperature variations affect the aggregation state and size distribution of SE structures.

In the case of the RLs, no differences were observed before or after the heat treatment for all the tested samples.

This behavior has already been reported in various studies, where RLs were found not to lose their activity even under extreme conditions at 121 °C [28]. Overall, these results confirm the remarkable thermal stability of RLs, which preserved their activity even after exposure to high temperatures, whereas SDS was more prone to aggregation due to its intrinsic temperature sensitivity. Thermal stability is a crucial indicator that enhances the applicability of RLs in fields such as food, medicine, and cosmetics, where cooking, sterilization, and pasteurization are commonly employed.

### 2.5. Emulsifying Activity

Different oil-to-water ratios were tested to determine the optimal conditions for each surfactant to function effectively. The results were analyzed considering the emulsion index (EI) and the droplet size of the generated emulsions. Following Bai and colleagues, we adopted an emulsification protocol more relevant to the food and cosmetic fields [4]. In the first emulsification test, conducted to identify the optimal oil-water ratio, M15RL was omitted, as the amount of purified material was limited. Emulsions were prepared using a surfactant concentration of 0.2 mg/mL. Preliminary emulsification tests at different concentrations were conducted during the initial screening phase to identify the most suitable working concentration. When an oil-to-water ratio of 1:1 was tested, none of the samples formed a stable emulsion, as indicated by the presence of a translucent oil layer on top of the test tubes (Appendix A). This behavior can be attributed to insufficient surfactant concentration relative to the oil volume, which was inadequate to stabilize the oil droplets, resulting in phase separation. In the case of the ratio 1:2.5, SDS had an EI value of 92.85 ± 1.11% followed by the M15CE with a value of 68.80 ± 1.72%; meanwhile, for SE, the lowest value was registered, 45.34 ± 16.57% (Figure 3).

In the case of ratio 1:5, the trend was confirmed, with SDS as the best emulsifying agent (90.23 ± 1.17%) followed by the M15CE and SE, respectively, 90.23 ± 2.97% and 85.88 ± 6.69%. The ratio 1:10 gave the best emulsions with the disappearance of the oil layer for almost all the samples. As shown in Appendix A, as the oil fraction decreases, phase separation becomes less evident and eventually disappears in all three samples at an oil-to-water ratio of 1:10, where all emulsions appear homogeneous (Appendix A). In this condition, SE registered the lowest emulsion index (EI) value (91.50 ± 1.03%) due to the presence of a thin cream layer on the top of the tube. It was hypothesized that the phase separation observed at higher oil fractions was primarily due to gravitational forces acting on emulsion droplets rather than surfactant inefficiency. For droplets larger than 1000 nm, Brownian motion is insufficient to maintain dispersion.

The analysis on emulsion droplets size gave remarkable results (Figure 3). In the case of SDS, for example, when the oil-water ratio was 1:2.5, the sample had the highest emulsion index but at the same time was the sample with the largest emulsion droplet size (2.22 ± 0.08 µm). In contrast, M15CE and SE, which had lower EI, showed an average emulsion droplet size < 2 µm. With the increasing addition of water (*o*/*w* ratio of 1:10), SDS recorded a significant decrease (*p*-value = 0.0069) in the dimension of emulsion droplets, with an average size lower than 1 µm (0.81 ± 0.02 µm). This size value is very similar to that obtained in the case of SE (0.87 ± 0.05 µm), which, however, was considered one of the samples with the lowest emulsion index even in the case of 1:10 oil-water ratio (91.50 ± 1.03%). Indeed, in the case of M15CE and SE, by changing the oil-water ratio, there was not a significant variation in emulsion droplet size, which remained below 1.5 µm in all the tested conditions (Figure 3). The droplet size distribution analysis (Appendix A) revealed a clear dependence on both the oil-to-water ratio and the type of surfactant. At the highest oil content (1:1), where emulsions exhibited phase separation, the droplet size distribution was strongly polydisperse with a small peak around 1 µm and a dominant peak above 100 µm, indicating the coexistence of small droplets with large aggregates (Appendix A). By decreasing the oil fraction, the distributions shifted progressively toward smaller sizes, with the main peak moving closer to 1 µm. At the lowest oil-to-water ratio tested (1:10), identified as the most stable condition, the distribution became narrower and more homogeneous, approaching a monomodal profile. However, even under these optimized conditions, a tail in the distribution remained, indicating the persistence of a minor fraction of larger droplets, particularly in the RL-stabilized sample. This observation is consistent with the bimodal character seen in the droplet size distribution (Appendix A) and with the optical microscopy images of the emulsions at the 1:10 ratio (Appendix A). These images further confirm that the droplet population is mainly uniform and close to monomodal, with only a few larger droplets, especially in the RL samples. Interestingly, although smaller and uniform droplet size usually correlate with a more stable emulsion, a different trend was observed in this case, suggesting that surfactants have various ways of stabilizing oil droplets. SE-stabilized emulsions displayed the narrowest and most monomodal distributions, already evident at the 1:2.5 ratio (Appendix A). Nevertheless, these emulsions showed the lowest EI, indicating that although the droplets were more uniform in size, their resistance to creaming and phase separation was limited. In contrast, emulsions stabilized by SDS and M15CE were initially bimodal at a 1:2.5 oil-water ratio and only gradually evolved into broader, yet closer-to-monomodal, distributions at an oil-water ratio of 1:10 (Appendix A). Despite their wider and less symmetrical profiles, these systems maintained higher EI compared to SE. This suggests that the stabilizing efficiency is not dictated solely by droplet uniformity, but also by the interfacial properties imparted by the surfactants. SDS and M15CE likely provide stronger electrostatic repulsion among droplets or a more resilient interfacial layer. Similar behavior has already been reported in a previous study, which is justified by the fact that RL is a low-molecular-weight surfactant that generally forms a thin interfacial layer [29]. However, a strong negative superficial charge characterizes them, which in some cases is enough to maintain the stability of the emulsion. It is noteworthy that the values reported for EI in our study are generally higher compared to other studies [19,28]. This could be a consequence of the unique way our emulsions were prepared and the fact that the calculation of the EI is based on the visual interpretation of complex phenomena, which may vary depending on the operator. This novel approach to studying emulsions adds a fresh perspective to the field.

In summary, our results indicate that RLs generally had a high emulsion index, demonstrating similar emulsifying performances to the commercial counterparts. This finding highlights the potential of RLs as a viable alternative in emulsion stabilization.

### 2.6. Rhamnolipids Crude Extract and Purified Mixture Emulsion Stability

The stability of the generated emulsion using the 1:10 oil-water ratio was investigated for a longer period (1 week) using M15CE, M15RL, SDS, and SE. Very high EI values were obtained with all the samples (Figure 4).

Some differences were observed between the emulsions produced by M15CE and M15RL. The emulsions stabilized by the purified mixture exhibited better stability, resulting in a significant decrease in the EI after one week. In contrast, M15CE showed a higher and considerable decrease in EI to 85.50 ± 3.52%. This difference was also noticed when the emulsion droplet size was analyzed. In the case of the M15CE, instability phenomena were observed, with the emulsion droplet dimension significantly decreasing to 0.77 ± 0.01 µm after 24 h. In the case of M15RL, emulsions were more homogeneous. However, a significant decrease in the emulsion droplet size was observed after 24 h and one week, which can result from creaming and the accumulation of higher-density oil droplets in the upper part of the tube. The observed slight decrease in emulsion droplet size over time can be explained by the dynamic nature of surfactant stabilization, which is not irreversible [30]. Surfactants can absorb and desorb from the oil–water interface over time, especially in systems undergoing slow structural evolution. The same trend of emulsion droplet size reduction with time was also observed in the case of SDS. In the case of SE, instability was observed with a significant decrease in the EI value to 79.89 ± 2.89% after one week and a non-significant fluctuation in emulsion droplet size.

A destabilization phenomenon was observed when evaluating the emulsifying performance of RLs at acidic pH. As shown in Figure 5, a consistent but non-significant increase in emulsion droplet size was observed when the emulsions were produced by M15CE at acidic pH, also indicated by the considerable variation found among the various replicates.

A significant decrease in the EI value was also observed after 24 h when a layer of free oil was visible on the top of the tube. This phenomenon was less evident in the case of the emulsion produced by M15RL, where the average emulsion droplet size, although increased to a value larger than 1 µm, showed greater homogeneity, and no oil separation was observed at the top of the tube after 24 h. We hypothesize that this can be attributed to the presence of other compounds inside the crude extract that can react to the acidic pH, increasing aggregation and instability of the emulsion. An interesting result was obtained from the surface charge analysis, where unexpectedly, the surface charge was lower in the case of the emulsion produced at pH 7, moving from −24.33 ± 4.05 mV to −29.0 ± 2.80 mV for M15RL and from −23.90 ± 1.08 mV to −28.03 ± 1.79 mV for M15CE. This finding contrasts with a previous study [3], in which the charge of the emulsion produced at acidic pH was closer to neutrality. Another study reported that the presence of one hydrophilic headgroup in the mono-RL renders it more stable at acidic pH compared to di-RL [31]. The difference between the study of Bai and McClements may rely on their use of a mixture of mono- and di-RLs. The lower stability of M15CE was also confirmed, as it was impossible to register an EI value after one week (Figure 6), and the oil was completely separated from the aqueous phase. Meanwhile, in the case of M15RL, an emulsion layer was still visible after one week with an EI value of 64.85 ± 1.45%.

### 2.7. Efficiency of Coenzyme Q10 Encapsulation

Due to the promising emulsification potential of RLs, we decided to proceed with testing the M15CE emulsion containing an active ingredient. Q10 was selected due to its strong antioxidant properties, making it suitable for various applications, including the food, nutraceutical, and cosmetic sectors [28]. While Q10 is synthesized endogenously, its production declines naturally during the aging process. Furthermore, the intestinal absorption of dietary Q10 is hindered by its lipophilic nature and high molecular weight, which contributes to its limited bioavailability from food sources [32]. The emulsion was prepared using Gum Arabic, a well-known stabilizer and film-forming polymer. The performance of M15CE as an emulsifier was compared with the performance of SDS since the emulsification behavior of the two compounds was similar (Figure 7).

Emulsions prepared with SDS were characterized by an average emulsion droplet size of 0.56 ± 0.07 µm, while those formulated with 0.2% (*w*/*v*) M15CE had an average size of 2.2 ± 0.39 µm. After 24 h, no significant changes in emulsion droplet size were observed, either at room temperature or under accelerated stability conditions at 40 °C (Figure 7A,B). After seven days at 40 °C, emulsions stabilized with 0.2% (*w*/*v*) M15CE showed destabilization, with an increase in the droplet size to 5.52 ± 0.41 µm and a decrease in the value of the EI to 53.25 ± 4.4% (Figure 7B,D). In contrast, SDS-based emulsions exposed to the same conditions remained stable, showing only a minor increase in particle size (0.52 ± 0.07 µm) and maintaining a high value of EI of 90 ± 4.7% (Figure 7B,D).

This destabilization is also evident from the droplet size distributions over time at different temperatures, shown in Appendix A. Examining the complete distributions provides a more detailed interpretation of the system behavior, revealing the broadening of the size profile and the appearance of larger droplets for the biosurfactant-stabilized emulsions at 40 °C after one week, in line with the observed increase in average droplet size. SDS-stabilized emulsions maintained a consistent, monomodal, and symmetrical distribution throughout the measurement period, indicating high stability. In contrast, RLs-stabilized emulsions exhibited destabilization at 40 °C after one week, with a broader distribution and the appearance of larger droplets, confirming the loss of stability of these systems under these conditions.

The 0.6% (*w*/*v*) M15CE exhibited a similar performance to SDS, resulting in an emulsion characterized by droplet sizes below 2 µm and a stable EI over the observed one-week period (Figure 7A–D). Specifically, after 7 days, emulsions maintained a droplet size of 1.07 ± 0.18 µm and an EI of 90.6 ± 2.4% at room temperature, while at 40 °C, they showed a droplet size of 1.05 ± 0.15 µm and an EI of 72.5 ± 5.6%. A limitation of the present study is the lack of rheological measures. Viscosity analysis will be considered in future investigations to better correlate emulsifying capacity with flow behavior. Significant differences were observed between M15CE and SDS formulations when assessing the amount of surface-exposed Q10 in the emulsions (Figure 7E). Indeed, when selecting the proper RL concentration, M15CE proved to be more active for encapsulation than SDS, as the former was more effective in reducing the amount of surface-exposed Q10. A similar result was also observed when Q10 was encapsulated in zein-propylene glycol alginate-rhamnolipid ternary composite nanoparticles [33] an optimal M15CE concentration increased the encapsulation efficiency of the delivery system, while an RLs excess negatively affected the amount of encapsulated compound, likely due to interfering with the homogeneous synthesis of the system. To better contextualize the encapsulation performance of the biosurfactant investigated in this study, we compared our results with two recent formulations based on natural surfactants for Q10 delivery. Liposomal Q10 delivery systems were previously formulated with the aim to replace cholesterol with β-sitosterol, a plant-derived phytosterol [34]. The authors reported that β-sitosterol not only increased encapsulation efficiency but also improved oxidative stability and cellular antioxidant activity in Caco-2 cells. Under optimized conditions, liposomes containing β-sitosterol demonstrated superior physicochemical stability compared to those containing cholesterol, resulting in approximately 15% reduction in lipid oxidation and an 11% enhancement in antioxidant activity. In a separate study, Uner and co-workers formulated CoQ10-loaded micelles using various natural saponins, including Quillaja saponin and ginsenosides R0 and Rb1 [35]. These saponin-based micelles showed improved encapsulation efficiency and smaller particle size compared to synthetic Pluronic F127 micelles. Moreover, they enhanced cellular uptake, modulated epithelial–mesenchymal transition (EMT) markers, and demonstrated promising anticancer activity in vivo. Although the systems differ (i.e., emulsions in our study, micelles in saponin-based formulations, and liposomes in phospholipid-based approaches) all utilize natural surfactants that show promising potential to replace traditional synthetic surfactants. In our formulation, biosurfactants enhanced Q10 encapsulation efficiency but required higher concentrations to achieve stable emulsions.

## 3. Materials and Methods

### 3.1. Chemicals

Sodium dodecyl sulfate 10% *w*/*v* (SDS) was purchased from Chem-Lan (Zedelgem, Belgium). Sucrose Ester (SE) SP 50 (HLB of 11) was gently provided from Sisterna BV (Roosendaal, The Netherlands). Corn oil was from a local market. Coenzyme Q10 was purchased from ACEF (Fiorenzuola d’Arda, Italy), and Arabic gum from FARMALABOR (Assago, Italy).

### 3.2. Production of P. gessardii M15 RLs Mixture

The RLs mixture from *P. gessardii* M15 was obtained as described before [9]. Briefly, *P. gessardii* M15 was cultivated in liquid TYP medium (16 g/L bacto-tryptone, 16 g/L yeast extract, 10 g/L NaCl) at 20 °C for five days. The supernatant was then collected, filtered (0.22 µm filter), and extracted twice with ethyl acetate 1:1 *v*/*v*. The purified M15RL was obtained by fractioning the crude extract M15CE through solid-phase extraction (SPE) using different percentages of methanol (MeOH) in water as eluent. The fraction M15RL was obtained by eluting 80% MeOH. The RL mixtures were dissolved in MeOH at 1 mg/mL and analyzed through Liquid Chromatography—High-Resolution Tandem Mass Spectrometry (LC-HRMS^2^) [36] as described in the Appendix A.

### 3.3. Surface Tension and Critical Micelle Concentration

Critical Micelle concentration (CMC) was estimated by measuring the surface tension of the surfactant solution at different concentrations. Samples were prepared by dissolving surfactants at the desired concentration in deionized water and stirred for 4 h. The surface tension (ST) of each sample was then measured four times with a Tensiometer DY300 (Kyowa, Japan) using the Wilhelmy plate method [37]. The surface tension instrument was calibrated prior to measurements following the manufacturer’s instructions. Briefly, the plate was placed on the instrument, and the system was allowed to reach equilibrium. Calibration was then performed using two 0.4 g weights positioned on the sides of the plate. CMC was determined by plotting the ST value as the function of the crude extract concentration. CMC was calculated as the point of intersection between the line passing through the plateau, representing the concentration range where biosurfactant molecules saturate the air-water surface, and the line where the ST shows a linear decline [38].

### 3.4. Particle Size, Polydispersity Index, and Surface Charge

The size (Z Average) and Polydispersity index (PDI) of the produced micelle were measured by dynamic laser scattering (DLS) using a Malvern Zetasizer Nano ZS (Malvern Instruments, UK). Disposable polystyrene cuvettes were employed for size and PDI analysis, and the samples were diluted 1:2 using demineralized water. The zeta potential was measured using a folded capillary zeta cell. Each sample was measured thrice, setting the refractive index to 1.46 and the absorption index to 0.

### 3.5. Mass Spectrometry of M15CE and M15RL Mixtures

The RL mixtures were dissolved in MeOH at 1 mg/mL and analyzed through Liquid Chromatography—High-Resolution Tandem Mass Spectrometry (LC-HRMS^2^). The chemical profiling was acquired with a Thermo Scientific Q Exactive Focus Orbitrap mass spectrometer coupled to a Thermo Scientific Ultimate 3000 HPLC system equipped with a Hypersil C18 (RPC18) column (100 × 4.6 mm, 3 μm) (Thermo Fisher Scientific, Waltham, MA, USA). The RP18 was eluted at 25 °C at 200 μL/min with H2O (A) and CH_3_CN (B), both supplemented with 0.1% formic acid. The gradient was set as follows: 25–60% B for 5 min, 60–100% B for 30 min, and 100% B for 20 min. MS spectra were acquired in negative ion mode. HESI source parameters were set as follows: sheath gas flow rate to 45 units N_2_, auxiliary gas flow rate to 10 units N_2_, spray voltage to 3.2 kV, capillary temperature to 285 °C, and auxiliary gas heater temperature to 150 °C. The full MS were acquired in the scan range of 150–1000 *m*/*z* with a resolution of 70,000 and an AGC target of 1 × 10^6^. MS^2^ spectra were acquired in the data-dependent analysis mode at a resolution of 70,000 and an AGC target of 5 × 10^4^, setting three MS^2^ events after each full MS scan. HRMS^2^ scans were obtained with HCD fragmentation, using an isolation width of 2.0 *m*/*z*, and normalized collision energies of 20 and 30 units.

### 3.6. Influence of pH and Thermal Treatment

The influence of different pH conditions and heat treatment on the surfactant’s activity was investigated by measuring the ST, particles size, and surface charge. To this aim, commercial surfactants and RLs mixtures were prepared by dissolving the samples in distilled water at 0.2 mg/mL and stirring for 4 h. Since the RLs content in M15CE was estimated to be 40%, the crude mixture was dissolved in water at 0.5 mg/mL to ensure a comparable concentration. The influence of pH was evaluated at acid (pH 3), neutral (pH 7), and basic (pH 12), adjusting the pH of surfactant solutions with 4M HCl or 1M NaOH. The influence of temperature on the surfactant’s activity was evaluated by incubating the surfactant solutions for 1 h at 90 °C using a thermal bath coupled with a thermal probe. After the treatment, an aliquot of the RLs samples was extracted with ethyl-acetate 1:1 (*v*/*v*), dried, and analyzed by HRLC-MS^2^ as described above.

### 3.7. O/W Emulsion Preparation

Corn oil was used as dispersed phase for the oil in water emulsions (*o*/*w*), and four different *o*/*w* (*v*/*v*) ratios were evaluated, such as 1:1, 1:2.5, 1:5, and 1:10. The emulsions were tested in triplicate in 15 mL tubes by adding the calculated amount of oil and water, in which the surfactants were already dissolved at 0.2% (*w*/*v*) or 0.6% (*w*/*v*). A first homogenization step was performed using an Ultra-Turrax T25 (IKA, Staufen, Germany) equipped with an S25N-10G probe for 2 min at 13000 rpm. This first emulsion was then sonicated using an SH 213 G ultrasound generator (Bandelin, Berlin, Germany) for 2 min at 30% amplitude operating at 20 kHz.

### 3.8. Emulsion Stability

The influence of acidic pH on emulsion stability was evaluated for M15CE and M15RL at an oil-water ratio of 1:10 *v*/*v*. Emulsions were prepared as previously reported by adjusting the pH to 3 by adding 4M HCl to the aqueous phase where the BSs were dispersed. Emulsions at neutral pH were used as control.

### 3.9. Emulsion Characterization

#### 3.9.1. Emulsion Index Assay

The emulsification ability of samples was determined by measuring the emulsion index (EI%) [39]. Briefly, 2mL of the emulsions were placed in transparent test tubes and allowed to stand at room temperature for 24 h. Appendix A in the Appendix A shows the two possible phenomena observed after the emulsion production to render easier the data interpretation [18]. After the incubation EI% was calculated as follows:(1)$$EI\%=\frac{Height\;of\;the\;emulsin\;layer}{Total\;height\;of\;the\;liquid}\times \;100,$$
by measuring the height (cm) of the different layers generated in emulsion versus the total height of the liquid. Measurements were performed after 24 h (EI24%) and after 7 days to further investigate the emulsification activity.

#### 3.9.2. Emulsion Particle Size

The size of the particles was measured using a Mastersizer 3000 (Malvern Instruments, Worcestershire, UK). Measurements were performed using the following parameters: RI 1.4493, AI 0.01 obscuration 15%. Prior to measurement, the emulsion samples were gently shaken to redistribute the droplets. Aliquots were collected from the central portion of the tube, avoiding the cream layer at the top. Three measurements were performed for each emulsion, and results were expressed in terms of surface mean diameter (D [3;2]).

### 3.10. Coenzyme Q10 Encapsulation and Encapsulation Efficiency

Emulsions were prepared by dissolving Arabic gum at 20% (*w*/*v*) in demineralized water with M15CE or SDS at a concentration of 0.2% (*w*/*v*) (Table 2). Since the starting concentration of RL in the crude extract was 40%, M15CE was dissolved at a concentration that allowed the same content of surfactants in the solution as SDS. For the RLs, a higher concentration of 0.6% was also tested to enhance emulsion stability (M15CE_B). The oil phase was formulated to achieve a 90:10 *w*/*o* ratio, with Coenzyme Q10 (Q10) solubilized in corn oil at a concentration of 100 mg/mL. The emulsion was formed through a two-step homogenization process. First, a pre-emulsion was prepared using an Ultra-Turrax T25 (IKA, Schwabach, Germany) equipped with an S25N-10G probe for 2 min at 13,000 rpm, followed by sonication using an SH 213 G ultrasound generator (Bandelin, Germany) for 2 min at 30% amplitude operating at 20 kHz. The resulting emulsion was characterized for the particle size and EI at RT, and under accelerated stability conditions (e.g., 40 °C, 40% relative humidity (Memmert, Germany).

The amount of non-encapsulated Q10 in the emulsion was determined to evaluate the encapsulation efficiency indirectly. Free Q10 was extracted by mixing 2 mL of the emulsion with ethyl acetate. The solution was mixed for 1 min using a vortex, then centrifuged at 5000 rpm for 10 min to facilitate the phase separation, enabling the Q10 recovery from the supernatant. The quantification was performed using a spectrophotometer (LLG, Germany) by measuring the absorbance at 272 nm. A calibration curve was used to quantify the active ingredient. The amount of encapsulated Q10 was indirectly estimated by subtracting the percentage of non-encapsulated compound from the total theoretical Q10 in the formulation.

### 3.11. Statistical Analysis

All measurements were performed in triplicate, and the results were reported as mean value ± standard deviation. Statistical analysis was performed using two-way analysis of variance (ANOVA) to assess the effects of surfactant type and experimental conditions (e.g., pH or thermal treatment) on dependent parameters such as micelle size, emulsion index, and droplet size. The significance level was set to 0.05. The biological samples of M15CE and M15RL were unique, while all experimental measurements were performed in technical triplicates.

## 4. Conclusions

The interest in finding suitable and more sustainable substitutes for synthetic surfactants is constantly increasing. However, only a few studies have shown the efficacy of biosurfactants in systems like those applied in industry. This study demonstrates that RLs produced by *Pseudomonas gessardii* M15, a novel Antarctic strain, exhibit lower critical micelle concentration (CMC), reduced surface tension, and remarkable thermal stability. RLs showed emulsification performance comparable to SDS and superior to SE. Importantly, RLs proved to be highly effective in encapsulating lipophilic bioactive compounds, such as coenzyme Q10, achieving higher encapsulation efficiency and minimizing surface exposure, which suggests improved protection and controlled release. These findings highlight the potential of RLs from *P. gessardii* M15 as natural surfactants for industrial applications, including food, cosmetic, and pharmaceutical fields. Future studies should focus on optimizing formulations, assessing toxicity and biocompatibility in food and cosmetic systems, and developing scaling-up strategies to exploit their emulsifying and encapsulation capabilities fully. Given the lack of previous studies on *P. gessardii*, further characterization of this strain and evaluation of fermentation scale-up are also necessary to assess its potential for industrial applications.

## Figures and Tables

**Figure 1 biomolecules-15-01451-f001:**
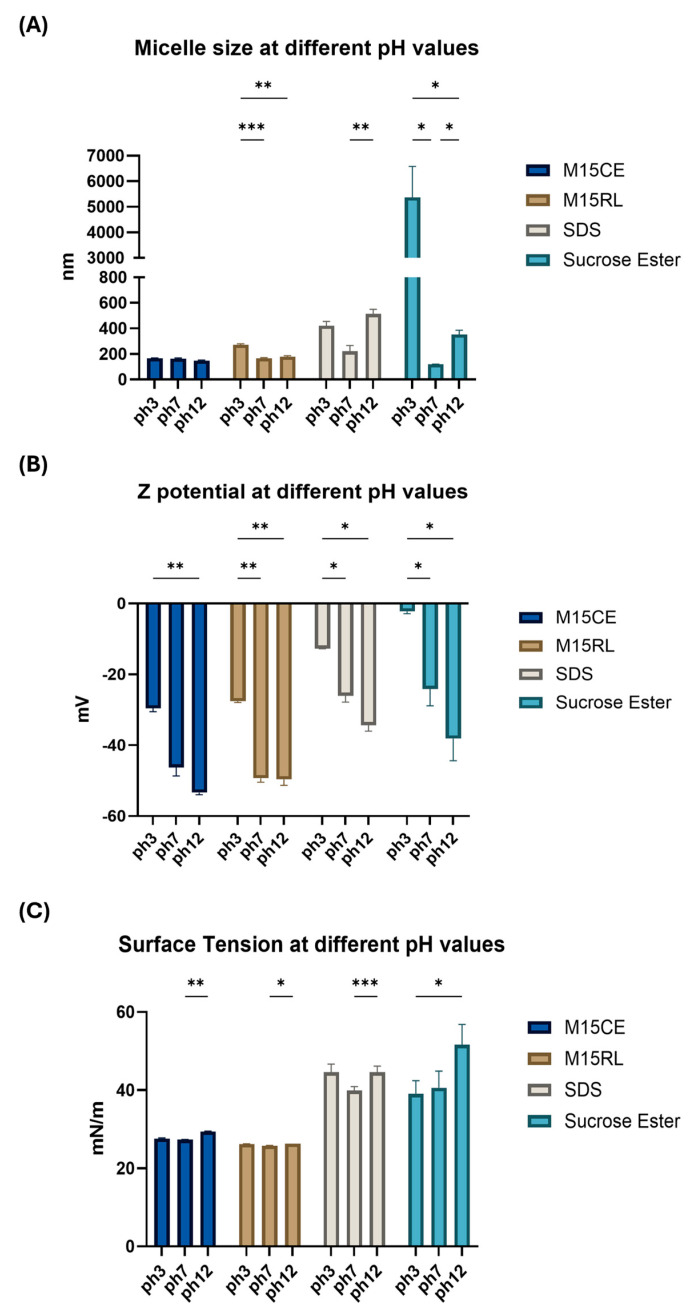
Characterization of micelle obtained at pH 3, 7, and 12. Surfactant concentration 0.2 mg/mL. (**A**) Micelle particle size, (**B**) surface charge, and (**C**) surface tension. Two-way ANOVA was utilized for statistical analysis. Tukey’s test was utilized for multiple comparisons. * *p* ≤ 0.05, ** *p* ≤ 0.01, *** *p* ≤ 0.001.

**Figure 2 biomolecules-15-01451-f002:**
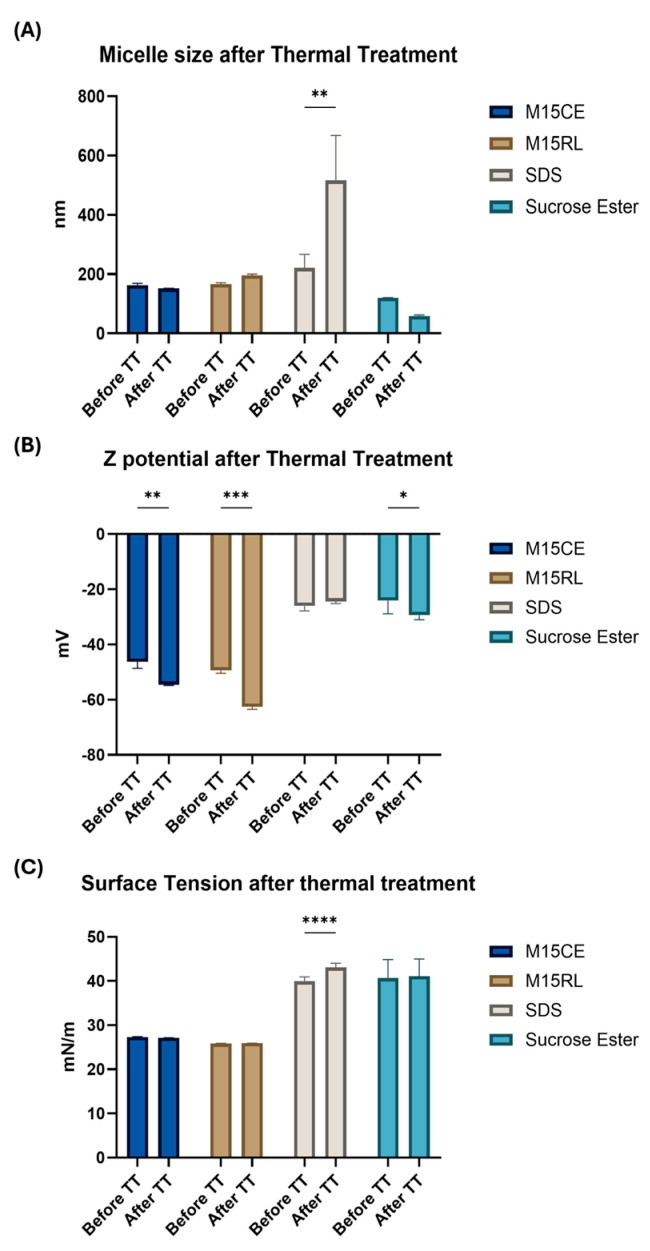
Characterization of the micelles following the heat treatment. Surfactant concentration 0.2 mg/mL. (**A**) Micelle particle size, (**B**) surface charge, and (**C**) surface tension. Two-way ANOVA was utilized for statistical analysis. Tukey’s test was utilized for multiple comparisons. * *p* ≤ 0.05, ** *p* ≤ 0.01, *** *p* ≤ 0.001, **** *p* ≤ 0.0001.

**Figure 3 biomolecules-15-01451-f003:**
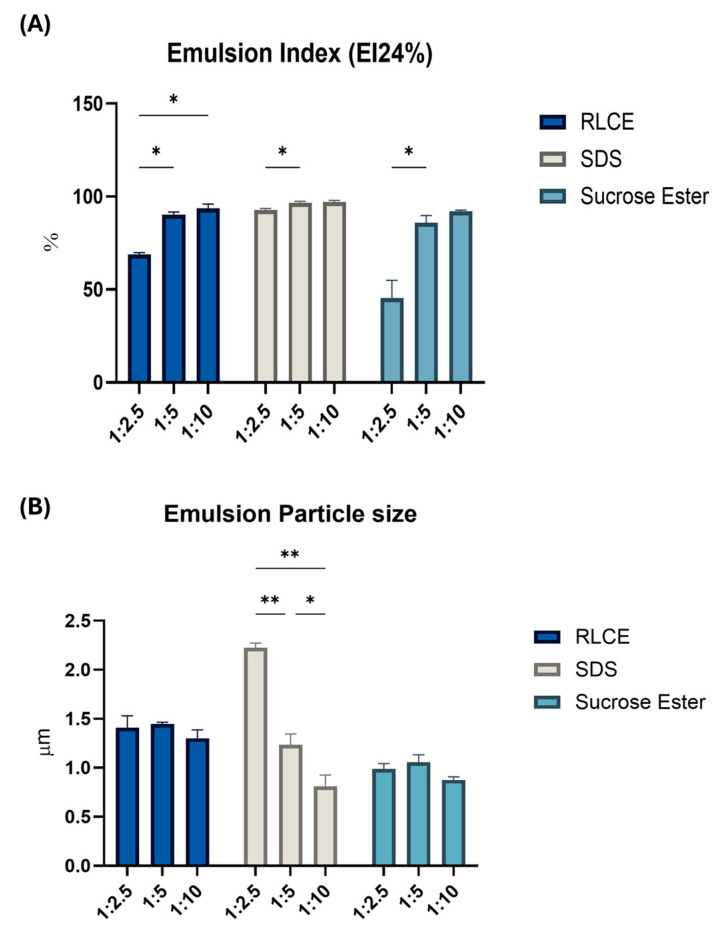
Comparison of the (**A**) emulsion Index (EI24%) and (**B**) emulsion particle size of M15CE, SDS and SE at different oil-water ratios. Surfactant concentration 0.2 mg/mL. Two-way ANOVA was utilized for statistical analysis. Tukey’s test was utilized for multiple comparisons. * *p* ≤ 0.05, ** *p* ≤ 0.01.

**Figure 4 biomolecules-15-01451-f004:**
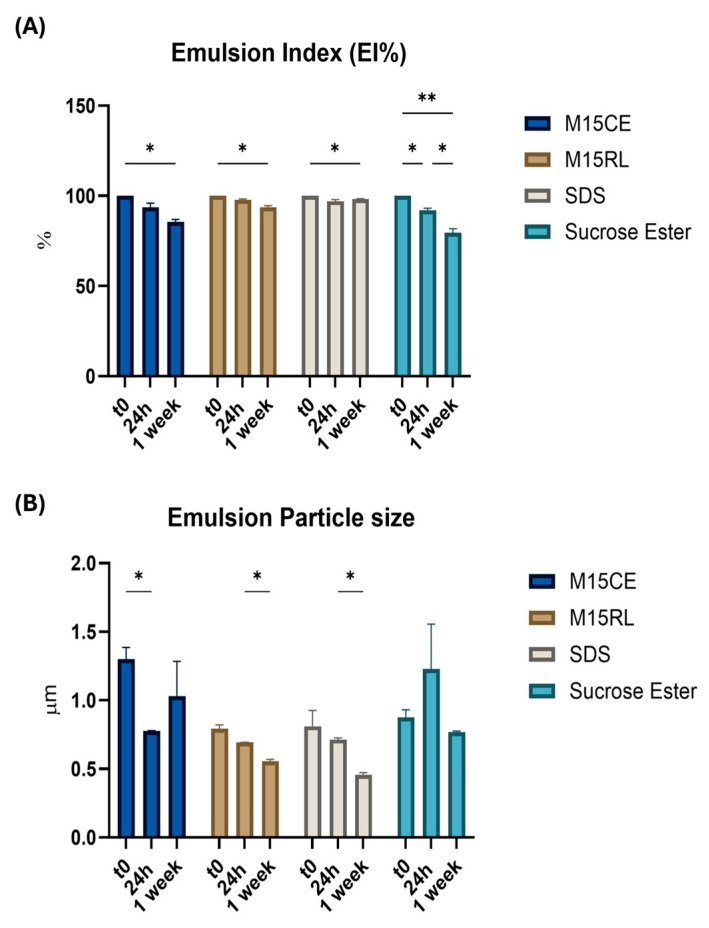
(**A**) Emulsion Index and (**B**) emulsion particle size evaluation after emulsification (t0), 24h and 1 week. Comparison between M15CE, M15RL, SDS and SE (concentration 0.2 mg/mL, 1:10 oil-water ratio). Surfactant concentration 0.2 mg/mL. Two-way ANOVA was utilized for statistical analysis. Tukey’s test was utilized for multiple comparisons. * *p* ≤ 0.05, ** *p* ≤ 0.01.

**Figure 5 biomolecules-15-01451-f005:**
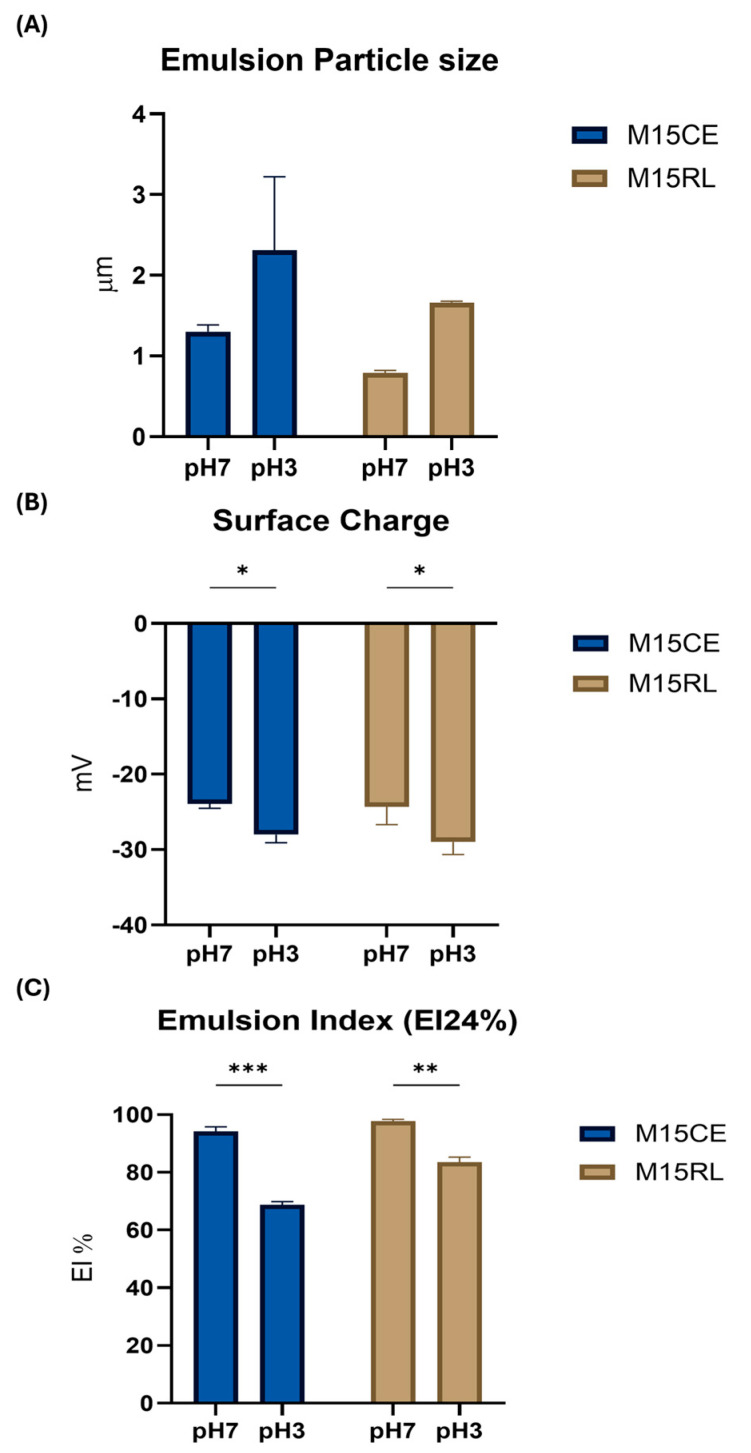
Comparison between M15CE and M15RL (**A**) emulsion particle size, (**B**) emulsion surface charge, and (**C**) emulsion index value after 24 h (EI24%) at pH 7 and 3. Surfactant concentration 0.2 mg/mL. Two-way ANOVA was utilized for statistical analysis. Tukey’s test was utilized for multiple comparisons. * *p* ≤ 0.05, ** *p* ≤ 0.01, *** *p* ≤ 0.001.

**Figure 6 biomolecules-15-01451-f006:**
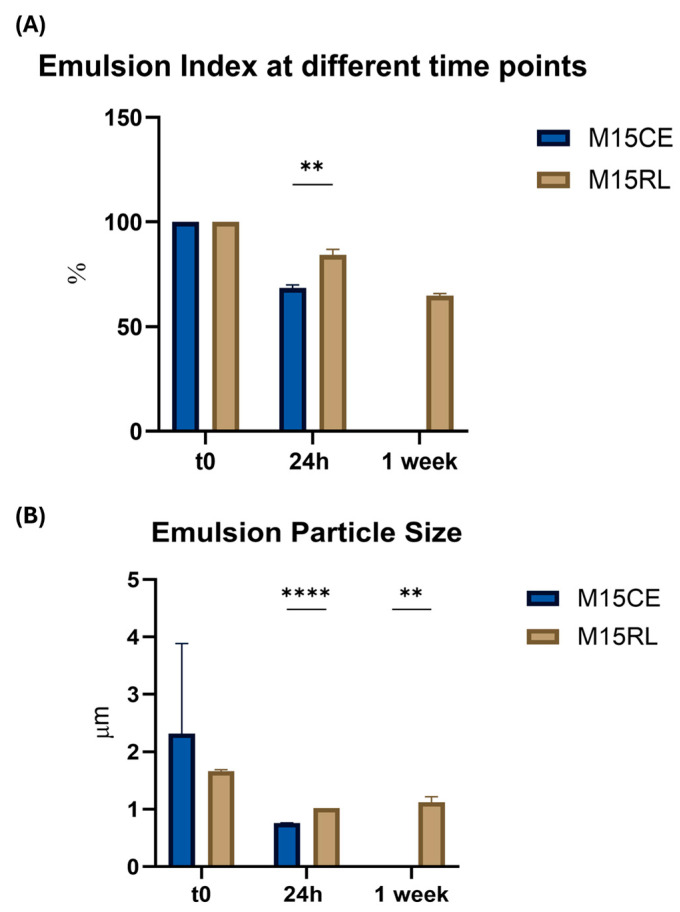
Comparison between M15CE and M15RL at pH 3. (**A**) Emulsion Index and (**B**) emulsion particle size evaluation after 24 h and one week. Surfactant concentration 0.2 mg/mL. Two-way ANOVA was utilized for statistical analysis. Tukey’s test was utilized for multiple comparisons. ** *p* ≤ 0.01, **** *p* ≤ 0.0001.

**Figure 7 biomolecules-15-01451-f007:**
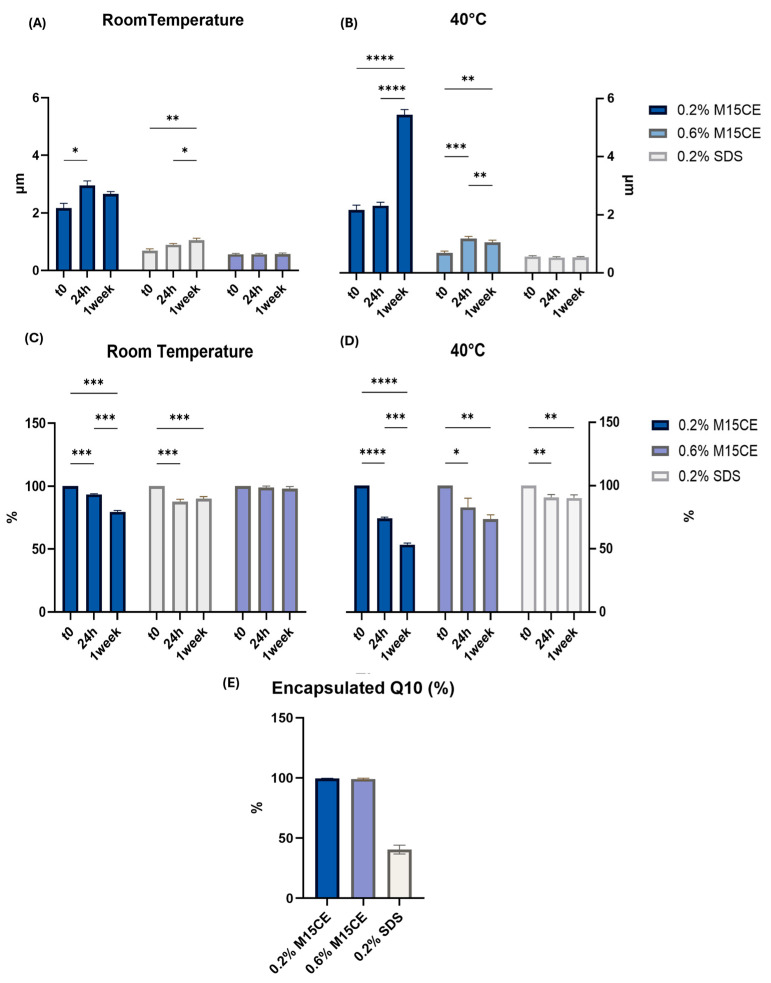
(**A**) Dimensional changes in Q10 encapsulated with 0.2% (*w*/*v*) M15CE and 0.6% (*w*/*v*) M15CE after 0, 24 h and 1 week at room temperature; (**B**) Dimensional changes in the different formulations over time at 40 °C; (**C**) Changes in the EI % at room temperature; (**D**) Changes in the EI % at 40 °C; (**E**) percentage of encapsulated Q10 for each analyzed formulation. Two-way ANOVA was utilized for statistical analysis. Tukey’s test was utilized for multiple comparisons. * *p* ≤ 0.05, ** *p* ≤ 0.01, *** *p* ≤ 0.001, **** *p* ≤ 0.0001.

**Table 1 biomolecules-15-01451-t001:** RL composition of M15CE and M15RL mixtures, and their relative abundances.

R_t_ (min.)	Rhamnolipids ^1^	Base Peak	M15CE (%)	M15RL (%)
16.14	Rha-C12:1	359.2075	0.5%	-
17.35	Rha-C12	361.2231	1.8%	-
18.92	Rha-C14:1	387.2388	0.8%	-
22.03	Rha-C14	389.2544	0.8%	-
23.61	Rha-C16:1	415.2701	0.1%	-
25.08	Rha-C8-C10	475.2907	10.4%	2.14%
27.59	Rha-C8-C11	489.3068	0.2%	0.21%
28.16	Rha-C9-C10	489.3066	0.7%	0.85%
28.77	Rha-C8-C12:1	501.3068	0.2%	0.47%
29.10	Rha-C16:1	417.2900	-	0.10%
29.37	Rha-C12:1-C8	501.3067	0.2%	0.51%
31.57	Rha-C10-C10	503.3223	22.8%	33.44%
32.82	Rha-C11:1-C10	503.3223	0.2%	0.81%
	Rha-C12:1-C9			
33.03	Rha-C12:1-C10:1	527.3224	0.1%	0.40%
33.82	Rha-C12:1-C10	529.3380	0.7%	2.39%
34.27	Rha-C11-C10	517.3380	0.7%	2.34%
35.15	Rha-C11-C10	517.3380	4.5%	9.49%
35.43	Rha-C10-C12:1	529.3379	1.1%	2.44%
36.43	Rha-C12:1-C10	529.3381	29.2%	27.43%
38.80	Rha-C12-C10	531.3538	17.6%	13.00%
39.55	Rha-C12:1-C12:1	555.3538	2.2%	2.08%
40.42	Rha-C14:1-C10	557.3694	2.6%	1.79%
41.20	Rha-C12-C11	545.3695	0.3%	0.22%
41.94	Rha-C12-C12:1	557.3694	0.9%	0.84%
42.17	Rha-C13-C10	545.3697	0.4%	0.39%
43.28	Rha-C12:1-C12	557.3694	0.4%	0.36%
45.20	Rha-C12-C12	559.3850	0.2%	0.18%
45.56	Rha-C14-C10	559.3851	0.2%	0.15%
45.99	Rha-C16:1-C10	585.4008	0.1%	0.10%
53.87	Rha-C16-C10	587.4166	0.1%	-

^1^ Rha is the abbreviation for rhamnose. Fatty acyl chains are indicated as Cn:x, where n is the number of carbon atoms, and x is the number of double bonds.

**Table 2 biomolecules-15-01451-t002:** Composition of the emulsions expressed as percentage relative to the volume of water. For surfactants, the reported values refer to the actual concentration of the active compound used. In the case of the crude extract (M15CE and M15CE_B), the concentration corresponds to the effective content of rhamnolipids (RLs) present in the extract.

Sample	Oil (%)	GA (%)	Surfactant (%)	Q10 (%)
M15CE	11.1	20	0.2	1.11
SDS	11.1	20	0.2	1.11
M15CE_B	11.1	20	0.6	1.11

## Data Availability

The original contributions presented in this study are included in the article/Appendix A. Further inquiries can be directed to the corresponding author.

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
