# Peer review of "Evaluation of the Emulsification Properties of Marine-Derived Rhamnolipids for Encapsulation: A Comparison with Commercial Surfactants"

_biomolecules, 2025, doi:10.3390/biom15101451_

Round 1

Reviewer 1 Report

Comments and Suggestions for Authors

This study explores the emulsifying and encapsulation performance of Rhamnolipids derived from the marine Antarctic bacterium Pseudomonas gessardii M15. Emulsification ability and stability were compared with the commercial surfactants Sodium dodecyl sulfate (SDS) and sucrose esters (SE) under extreme conditions of temperature and pH. These findings support its use as a sustainable alternative to synthetic agents in diverse industrial settings. The manuscript looks fine and may be published after following revision.

  1. Overall, English is looking fine and still more polish thoroughly.
  2. Introduction should be elaborate, as it looks little bit sort.
  3. Authors should provide the calibration procedure of surface tension instrument employed in current study.
  4. Authors should give the structure of M15CE and M15RL.
  5. Authors should provide viscosity study of employed system and compare their results.
  6. If possible, authors should also include foamability of surfactant solutions and then compare.

Author Response

Dear reviewer, Thank you for your time and willingness to comment on our article. In the attached file you will find our comments and responses, along with the indications of any additions in the revised version.

Best regards 

Francesca 

Reviewer 2 Report

Comments and Suggestions for Authors

The manuscript I reviewed, titled: “Evaluation of the emulsification properties of marine-derived rhamnolipids for encapsulation: a comparison with commercial surfactants” addresses the interesting topic of obtaining more biocompatible and well-tolerated surfactants. However, I find certain gaps in the broader understanding and potential applications of the proposed compounds. The materials and results sections also require some supplementation. Detailed comments are provided below.

  1. The introduction lacks more detailed data on the potential use of these compounds and their future technological potential (what applications, formulations, routes of administration?). This is important from the perspective of considering their properties for future applications. I expect authors with expertise in this topic to provide a concise yet substantive presentation of key data on this topic.
  2. I also lack a concise but substantive characterization of the surfactants selected for comparison (SDS and sucrose esters).
  3. Figure 1A – what accounts for such a high value (in µm) of micelles at pH 3 with sucrose esters? Is this an artifact?
  4. Figure 1: What is the justification for using three significance levels (* p≤0.05, ** p≤0.01, *** p≤0.001)?
  5. Figure 3B particle size vs. Figure S1 – is phase separation is not a result of droplet size (>1000 nm) and not just surfactant effectiveness? At this droplet size, Brownian motion does not stabilize the system. Please add a comment to the manuscript on this topic. Is the translucent layer at the top of the tubes in Figure S1 the aqueous or oil phase? There is no clear indication of which systems showed phase separation—i.e., emulsion breakage. Please add this information.
  6. Item 3.4.: In the DLS measurement, what sample was diluted with water? Was methanol present? If so, what effect did it have on the obtained results?
  7. 3.9.2. Emulsion particle size: The mean value D was given as the characteristic parameter. Was there a visible effect of surfactants on the particle (droplet) size distribution? Were the distributions mono-, bi-modal, symmetrical, or asymmetrical? This is a very important aspect, so when conducting this type of study, the obtained results should be discussed in more detail than just the mean value.
  8. Point 3.10.: Please provide a table with the exact percentage composition (w/w) of the tested systems, so that the concentration of oil, water, coenzyme Q10, and the tested surfactants in the individual mixtures is clearly visible.
  9. Point 3.11.: Please indicate and justify the selection of tests performed in the statistical analysis.
  10. Lines 453-454: Please specify what "various commercial preparations" means.
  11. Additional materials: Figure S1 suggests a creaming process (top layer) in emulsions, while Figure S2 shows the complete opposite process in emulsions, not creaming but rather sedimentation. Please explain this and change Figure S1.
  12. Additional materials: Figure S2 has an incomprehensible caption, four vials in each figure (a-d), and three surfactants (M15CE, SDS, and SE). Please explain.

Author Response

(The authors gave the same response as above.)

Reviewer 3 Report

Comments and Suggestions for Authors

The study addresses an important area in biosurfactant research, focusing on marine-derived rhamnolipids (RLs) as sustainable alternatives to synthetic surfactants.

The selection of 0.2 mg/mL for testing is justified, but it would strengthen the work if the authors provided a concentration-dependent emulsification analysis (not only fixed-point comparisons).

Some details on experimental replicates are given, but statistical robustness should be emphasized (e.g., clarify number of biological vs. technical replicates).

Figures 1–2 (micelle size, zeta potential, and surface tension at different pH and thermal treatments): The interpretation could be expanded, especially regarding why RLs outperform SE in stability yet form larger micelles.

Figure 7 (CoQ10 encapsulation): This is the most compelling dataset, showing much higher encapsulation efficiency. However, comparisons with other natural surfactants (e.g., saponins, phospholipids) would help contextualize the real-world relevance.

The Introduction is informative but too descriptive. It should more directly identify the research gap.

The Conclusion should avoid redundancy and instead highlight future directions (e.g., application in food/cosmetics, formulation optimization, toxicity/biocompatibility studies).

Supplementary materials (Table S1, Figures S1–S2) contain crucial comparative data—authors should ensure these are properly cited in the main text.

Some references are dated; more recent work on biosurfactant encapsulation (2022–2024) should be integrated

Author Response

(The authors gave the same response as above.)

Reviewer 4 Report

Comments and Suggestions for Authors

The present paper explores the surface, bulk and emulsification properties of rhamnolipids (RL) derived from the marine Antarctic bacterium Pseudomonas gessardii M15. This topic is interesting as currently there is a high demand to replace the synthetic surfactants with natural or bio-surfactants. The authors synthesize the studied rhamnolipids, characterize their composition using LC-MS and compare their properties with those of two other commercial surfactants – the anionic sodium dodecyl sulfate (SDS) and the sucrose ester SP50 (denoted as SE). They conclude that the properties of the studied RL are comparable or even superior than those of SDS and SE. I have the following comments and questions:

Major comments

  1. Results with SDS. Figure 1 shows results for SDS “micelles” presumably measured at 0.2 mg/ml concentration (as explained in the text, line 118 or at 0.067 mg/ml if 1:2 dilution has been applied as suggested in Section 3.4).

However, 0.2 mg/ml = 0.2 g/l = 0.694 mM (SDS Mw = 288.38 g/mol).

Even this concentration is more than 1 order of magnitude lower compared to the CMC values for SDS known from the literature ≈ 8.2 mM ≈ 0.2 wt. % = 2 mg/ml, see e.g. Moroi et al. JCIS 1974 46 111-117 (CMC reported to be 8.25 mM at 30°C); Markarian et al. J. Sol. Chem. 2005 34 361 (CMC reported to be 8.2 mM at 25°C); Nivon-Ramirez et al. J. Colloid Surf. A 2022 645 128867 (CMC calculated via MD simulations ≈ 8.25 mM at 25°C).

Furthermore, the conclusion that there are no micelles in the studied solutions is further supported by the quite strange results reported by the authors – micelles with size of 220 nm are reported for SDS. However, it is very well known that at low concentrations, the SDS micelles have size of about 4-6 nm as reported by numerous other studies (see e.g. Schafer et al. Angew. Chem. Int. Ed. 2020 59 18591; Duplatre et al. J. Phys. Chem. 1996 100 16608; Migorod et al. Chem. J. Moldova 2019 doi: 10.19261/cjm.2019.572).

Therefore, it remains unclear what exactly the authors have measured. Furthermore, it remains also unclear how the CMC of SDS reported to be 0.142 mg/ml (line 108) has been determined.

The results obtained after thermal treatment also need further explanation.

  1. Results with SE. Figure 1 further show the “micelle” sizes from the SE solutions. However, although here the reported results per se may be correct, their interpretation is not, since spherical particles with sizes of 120 nm are clearly not micelles. Instead, most probably the authors have observed the size of the diester particles formed upon the SE dissolution, as previously shown by Pagureva et al. JCIS 2024 674 209-224. Furthermore, the changes in the sizes shown upon pH change are in good agreement with the results shown in Cholakova et al. JCIS 2025 693 137610. However, there is significant difference between micelles and SE particles, which the authors should properly discuss and explain.

Furthermore, the explanation of the negative zeta potential for the SE “micelles” (lines 145-148) should be also revised, accounting that the SE are in principle nonionic surfactants and no charge is expected in general.

  1. Study design. The authors should explain why double-chained RL surfactants need to be compared with single chained C12-anionic surfactant and a mixture of single- and double-chained nonionic SE surfactant. This choice is quite strange and unjustified. Furthermore, no any comparison is made with rhamnolipids derived by other sources, which seems much more appropriate to highlight the advantages/disadvantages of the presently studied RL. Such comparison needs to be included in the paper, as well as more detailed comparison for the surface/bulk and emulsification properties of other RL previously studied in the literature.
  1. Lines 247-249 “a significant decrease in the emulsion size was observed after 24h and one week, which can result from the creaming and accumulation of higher-density oil droplets in the upper part of the tube”.

This sentence contains several conceptual flows:

  • All oil droplets have the same density, unless they have different compositions which is not suggested by the authors
  • Bigger in size drops will cream quickly compared to the smaller in size droplets. However, if no coalescence is observed – then the average emulsion drop size will not change. Simple re-homogenization should be sufficient to redistribute all drops homogeneously throughout the sample. However, if this has not been done by the authors – then the results about the drop sizes are compromised.
  1. Conclusions section

All conclusions need to be justified by the presented results. The authors claimed for example that the studied RL minimizes “the amount of surface-exposed coenzyme Q10”. However, this has not been shown. Furthermore lines 452-453 suggest that M15 RL have “antibacterial and antiviral activities” which is also not shown in the current study and not references are provided. The comparison with the commercial surfactants need to be made in a proper way.

Other comments

  1. Introduction section needs to be revised and previous results with RL should be included. Lines 54-55 needs to be revised; Refs. 5 and 7 do not seem appropriate for the purposes they have been used; Some references are missing, e.g. in line 65
  2. Table S1 – do not show the percentage content of different RL in the studied samples. This should be revised. Furthermore, the main species in the studied RL should be disclosed also in the main text. The differences between M15CE and M15RL should be explained clearly in the main text.
  3. Surface tension measurements – surface adsorption isotherms (Section 2.2) should be included in the Supplementary Materials. Furthermore, the obtained results should be used to calculate the adsorption parameters – adsorption, area/molecule… These parameters should be taken into account for the appropriate design of the emulsion experiments, accounting what would be the needed amount of surfactant to produce emulsion with a given average drop size with respect to the oil content. Instead of comparing various surfactants at the same concentration, the comparison needs to be performed at the same level as compared to the amount needed to cover the drop surface.
  4. Lines 102-103 – how is the HLB of the M15 RL assessed?
  5. Concentrations of all surfactants needs to be clearly stated in all figure captions.
  6. Line 180 – “solubilization” is suggested – this needs to be explained/revised.
  7. Figure S2 – 4 tubes are shown in each pictures but only 3 surfactants are stated to be tested in the figure caption.
  8. Average “particle” sizes – Figure 3 – optical microscopy pictures needs to be provided. Given that in most of the emulsions bulk oily layer is seen in Figure S2 – it is highly probable that the drops are much bigger and DLS is not an appropriate technique to determine accurately the drop sizes in the prepared emulsions.
  9. Discussion about the drop sizes - needs to be revised, see e.g. “micelle dimension with an average size lower than 1 um (0.81 um)” – these are not micelles but emulsion droplets! Furthermore, emulsion droplets are also called particles (line 190), oil particles (line 219) which is not entirely correct.
  10. Line 224 – further details are needed. Lines 225-227 need to be revised. Conclusion of part 2.5 should be convincingly supported by the presented experimental results. Further results demonstrating the emulsions appearance and drops observed under an optical microscope with and without sample dilution are needed.
  11. Fig. 4what is the reproducibility of these results?
  12. Lines 268-272 – are these zeta potentials within the experimental accuracy. If not – then an explanation or potential mechanism explaining them should be provided.
  13. Lines 315-319 – further explanations, proofs of these conclusions need to be provided.
  14. “Demineralized water” – are the authors meant “deionized” or some of the ions remained in the water used in the present experiments? How was this water purified?
  15. Section 3.7 – the surfactant concentration needs to be included.
  16. Line 400 “negative control” – why?
  17. Line 410 “emulsification activity” – what did the authors meant by that?

    Any papers recommended in the report are for reference only. They are not mandatory. You may cite and reference other papers related to this topic. 

Author Response

Dear reviewer, Thank you for your time and willingness to comment on our article, your suggestions were truly appreciated. In the attached file you will find our comments and responses, along with the indications of any additions in the revised version.

Best regards 

Francesca 

Round 2

Reviewer 1 Report

Comments and Suggestions for Authors

Accept in its current form.

Reviewer 2 Report

Comments and Suggestions for Authors

Thank you for answering all my questions and takiego my comments i to account in the manuscript.

Reviewer 3 Report

Comments and Suggestions for Authors

manuscript is improved